# Heat Stress and an Immune Challenge Influence Turkey Meat Quality, but Conspecific-Directed Pecking Behavior Does Not

**DOI:** 10.3390/foods11152203

**Published:** 2022-07-25

**Authors:** Melissa Davis, Rachel Stevenson, Emily Ford, Marisa Erasmus, Stacy M. S. Zuelly

**Affiliations:** Department of Animal Sciences, Purdue University, West Lafayette, IN 47906, USA; davis565@purdue.edu (M.D.); steve189@purdue.edu (R.S.); ebedwell@purdue.edu (E.F.); merasmus@purdue.edu (M.E.)

**Keywords:** pre-slaughter stress, heat stress, immune challenge, turkey, meat quality, behavior, pecking

## Abstract

Heat stress (HS), immune challenges (IC) and pecking behavior are some of the many stressors poultry can experience in commercial settings that may affect bird welfare and meat quality after harvest. The first objective was to determine if HS or IC turkeys displayed greater negative effects on meat quality, and the second objective was to determine if the frequency of non-aggressive pecking behaviors among the birds was related to meat quality. Ninety-two, commercial male, beak-trimmed turkeys were used with a total of 15 rooms and 4–7 birds per room. Each treatment was applied for 1 week prior to harvest: the Control (CON) group had no stressors added, the HS group ambient temperature was approximately 29 °C for 120 min, and the IC group involved inoculating birds with a live vaccine for hemorrhagic enteritis virus. Birds were recorded and scored to quantify pecking behavior. Once harvested, carcasses were evaluated for feather retention force, pH, color, proximate analysis, fatty acid composition, shear force, and drip loss. Stress treatment resulted in HS breasts having the lowest protein content, and IC breasts having the lowest CIE L* values and the greatest shear force values. Pecking behavior had no impact on any meat quality attributes.

## 1. Introduction

There are many stressors that affect the final meat product quality of commercially raised food animal species. A stressor is an agent that produces stress for an animal and stress is the animal’s individual biological response to that stressor that disrupts the animal’s normal behavioral activities [1]. Environmental stressors such as heat stress or disease have become major points of interest to producers and consumers in recent years, and poultry species are very susceptible to these types of stress [2,3,4,5].

Heat stress is a significant economic and welfare issue within the turkey industry. Another stressor that many species face is the idea of social stress in group living situations. Stress in groups can result from aggressive or non-aggressive interactions. Non-aggressive interactions, such as feather pecking, can lead to pain, feather loss and damage and in some cases, to cannibalism; feather pecking is the repeated plucking of feathers of another individual and can be categorized as an investigative or modified foraging behavior [6,7]. Aggressive interactions, on the other hand, are used to maintain the social dominance hierarchy, but can also lead to stress and decreases in animal welfare. Social hierarchies are designed so that each member understands its role or place within the group relative to the other members, and then “rules” are created to control social encounters with one another [8]. A social hierarchy is also unique to each group, and will change with the addition or removal of an individual [9]. Male turkeys are particularly aggressive towards one another, and the advantage of being the highest-ranking individual in the group is priority access to resources [7,10].

Aggressive interactions among turkeys are typically defined as forceful, repeated pecking bouts directed at the head or body of another individual [6,7]. Craig [11] discussed how the behaviors that poultry species exhibit within their social hierarchies could have an effect on both their well-being and production characteristics such as meat quality. Considering there is very little research on this subject, especially in turkeys, the possibility of a relationship between social status and meat quality should be assessed. Studies since the early 1920s have shown positive relationships between social rank and production traits, specifically in egg quantity and quality [12,13,14,15]. However, these studies have only gone as far as to measure weight gain and final body weight, and we are unaware of any research on the possible relationship between the social ratio of turkeys and their fresh meat quality attributes since most studies similar to this have been done on chickens.

Over the last several years, research has increased on the effects of stress conditions on meat quality [16]. Meat quality is defined by the compositional quality and the palatability factors such as appearance, tenderness, flavor, and juiciness. Poor animal welfare can lead to poor product quality and several conditions in meat such as abnormal meat color, pale soft and exudative meat (PSE), dark firm and dry meat (DFD), poor shelf life, bruising, torn skin, and broken bones (Gregory and Grandin, 1998). Several studies in the last few years have shown impaired growth performance in poultry species subjected to periods of heat stress [17,18,19,20,21]; however, we do no’t know as much about welfare and meat quality in turkeys as we do about them in other poultry species. Furthermore, environmental stress factors can result in changes in the metabolites of muscle that are responsible for the differences in the ultimate properties of meat [22].

Another quality factor that has been a problem in the industry is feather retention force, or the force required to remove a feather from the feather follicle. Handling and treatment prior to slaughter can influence the force required to pluck the feathers from their carcasses during harvest [23]. Carcasses with difficult feathers to pluck will slow down production lines and ultimately cause a decrease in economic output. There is little research on stressors affecting feather retention force; however, it is important to know what factors affect feather retention force in order to minimize it.

Currently, there have been several studies examining the impact of heat stress on meat quality and immune function in other species; however, there have been no studies on a combination of heat stress and vaccinated immune stress at different age points in turkeys. We hypothesize that heat stressed, socially stressed, and vaccinated turkeys would show negative effects on fresh meat quality and feather retention force (FRF) compared to turkeys in the control group. Therefore, the aim of the present study was to determine if the heat stressed birds or immune stressed birds showed greater negative effects on overall meat quality compared to a control group, and to understand how stimulation of a turkey’s immune response through vaccination may influence meat quality. We also wanted to determine if individual differences in pecking frequency influence turkey fresh meat quality attributes.

## 2. Materials and Methods

### 2.1. Animals and Housing

This study involved two trials (in-time replicates) with a total of 92 turkeys (Trial 1: 50 turkeys, Trial 2: 42 turkeys). The commercial male beak trimmed turkeys (Nicholas Select, Aviagen Turkeys, Lewisburg, West Virginia) were received from a commercial hatchery at 1 d of age to the Purdue Animal Sciences Research Center (ASREC). From 1 to 7 days of age, the turkey poults were housed together in a brooding ring, and then randomly assigned to 8 littered (wood shavings) pens (measuring 2.44 m by 1.52 m) in Trial 1 and 7 pens in Trial 2, with 4 to 7 birds per pen for both trials.

Each pen included a hanging feeder and bell drinker to provide feed and water ad libitum. Turkeys were fed starter, grower and finisher diets that were formulated to meet breeder recommendations [24]. Room temperature and lighting were maintained according to industry standards [25]. For 1 day, poults were given 24 h of light that was gradually adjusted to 15 h light: 9 h darkness by the fourth day. A minimum light intensity of 40 lux was maintained, and room temperature was changed weekly as recommended by Aviagen [25]. Poults were brooded at a temperature of 30 °C, which was gradually adjusted to a final temperature of 13 °C by week 14. At 7 weeks of age, the turkeys were moved to the Purdue University Veterinary Animal Isolation Building (VA2). Birds that were housed together previously continued to be housed together in the new building. Each room had its own separate temperature and lighting controls, but each room was set to a lighting schedule of 0700 to 2100 with an average temperature (±SD) of 17.9 ± 1.9 °C.

At 7 weeks of age, the turkeys were individually marked with a black non-toxic livestock marker (Prima Tech Marking Stick, Neogen Corp., Lansing, MI, USA) to be able to identify each bird in each room. The livestock marker was reapplied every two weeks until harvest to ensure the markings did not fade.

Temperature and humidity sensors were placed in each room at the height of the turkeys for an accurate reading. To examine activity levels of turkeys, accelerometers (AXY-3 Micro Acceleration Data Loggers, TechnoSmArt, Guidonia Montecelio, Italy) were attached to one leg of two birds per room using a Vet Wrap bandage. Turkeys assigned to wearing the accelerometers were familiarized to the Vet Wrap wrapped around their leg 1 week prior to the data collection week.

### 2.2. Treatment Experimental Design

This study was part of a larger study intended to examine the influence of heat stress and an immune challenge on the welfare and activity levels of turkeys. Therefore, the experiment was designed to examine the behavior of individuals under different circumstances (heat stress, immune challenge) so that individuals could be used as their own controls, while also comparing the effects of the treatments among groups of turkeys. Each room experienced each treatment, but treatments were applied in different orders at 10, 12 and 14 week of age (Table 1). The treatment groups included a heat stress (HS), an immune challenge (IC) and a control condition where no other treatment was given (CON). Between the two trials, five total rooms (*n* = 5) were randomly assigned to each of the three treatment orders (Table 1).

During the HS treatment days for each time point, the room temperature gradually increased until the rooms reached a peak temperature range that depended on each room’s humidity level (Table 2). In order to determine the correct temperature range to use, a heat index chart that was created for hen turkeys was referenced from Xin and Harmon [26]. To ensure that the heat stress was mild, a heat index reference chart [26] was used to confirm that the temperature ranges chosen were defined as up to “danger” but did not reach temperature ranges defined as “extreme” heat stress. The humidity of the rooms was also recorded and applied to the reference chart to help determine the temperature range. To verify that the turkeys experienced heat stress, cloacal body temperatures were recorded for two turkeys in each of the HS rooms (Table 3).

Each of the rooms took approximately 140 min to reach the peak temperature range and the temperature was held there for 120 min. Then, the rooms took approximately 140 min to cool down to the temperature the room was originally at. During the heating and cooling process for each HS room, an observer was recording the temperature and humidity levels every 10 min to determine when the peak temperature range for the heat stress was reached.

The immune challenge treatments were given using a live-virus hemorrhagic enteritis vaccine (Oralvax HE^®^, Merck Animal Health, Rahway, NJ, USA) that was administered via the drinking water. The vaccine was prepared according to the manufacturer’s directions. The vaccine was prepared in a 3.785 L jug by pipetting 0.375 mL of the rehydrated vaccine into the jug. Depending on how many birds were in each room (4 to 7 birds), only the amount of vaccine water was prepared for each pen. On the day of the IC treatment, the bell drinkers were removed from the rooms 2 h prior to the vaccine administration to ensure that all birds would drink the vaccine water when the drinkers were returned. The water containing the vaccine was provided after the 2 h deprivation period and then was left in the rooms for 2 h before it was replaced with fresh, clean water. A timeline was created to outline the schedule of treatments for each treatment day (Table 4).

Turkeys in CON treatment groups were not subjected to any heat stress or immune challenge. For analysis of the data, the last stress treatment imparted 1 week prior to harvest on the pen is the designated treatment group.

### 2.3. Turkey Behavior

An overhead camcorder (Sony Camcorders, CX405, Sony Corporation of America, New York, NY, USA) was installed into each room in order to monitor the turkey behavior. The roles (Table 5) and behaviors (Table 6) of the turkeys were analyzed from the video recording two days prior to treatment at 14 weeks of age. Each turkey was observed continuously for 15 min at 1300 and 1600 h two days before the treatments were imposed.

### 2.4. Non-Aggressive Pecking Frequency Determination

Methods for determining the social ratio amongst the birds was completed using the outcome of all agonistic interactions between pairs of birds. A bird was considered the instigator if observed to be giving the pecking, and the recipient if that bird was receiving the pecking. The values were determined for the birds according to which other birds they dominated within the periods of time observed (adapted from Leonard and Weatherhead [27]). A calculation for each bird’s behavior was used to determine an aggressive (aggressive pecks), non-aggressive (feather pecking or beak pecking) and approach/avoidance (chase, threat, avoidance) values to compare to the other birds in the same room (adapted from Cordiner and Savory [28]). This ranking method is similar to a “dominance ratio” or “social tension index” that is used for characterizing agonistic behaviors of birds in flocks without actually calculating the pecking order, and this has been found to be highly correlated with peck order in a previous study with laying hens [28]. Cordiner and Savory [28] explained that the ranking orders developed using this method reflect the relative propensities of individual birds to give or receive different sorts of pecks, and thus indicate their status in different contexts. Calculations were done this way instead of using pair-wise “fights” since placing birds in pair-wise encounters may in turn create a dominance relationship rather than measuring one already established in the group [29]. Calculations were as follows:

For Aggressive Interaction Value Per Bird = (All aggressive pecks given + 1)/(All aggressive pecks received + 1)

For Non-Aggressive Interaction Value Per Bird = (All non-aggressive pecks given + 1)/(All non-aggressive pecks received + 1)

For Approach/Avoidance Interaction Value Per Bird = (All approaches + 1)/(All avoids + 1)

One (1) was added to both the numerator and denominator values to ensure that the resulting values were never zero. If a bird resulted in a value of 1, then they received as many pecks or approaches as they gave. If a bird had a value of <1, then they received more pecks or approaches than they gave, and a value of >1 meant they would give more pecks and approaches than they received.

The behaviors of calculated interactions per bird per hour were used to determine if there were sufficient data in all behavior categories (non-aggressive, aggressive, and approach/avoidance) to use them as indicators of social dominance. The instances of aggressive interactions and approach/avoidance interactions did not occur often enough (<1 interaction per bird per hour); therefore, only values for non-aggressive interactions such as feather pecking and beak pecking were used to determine a non-aggressive pecking ratio. The values were then divided by the number of birds in each room to standardize each turkey’s ratio for the number of turkeys in the room. Using the PROC FASTCLUS procedure in SAS (SAS version 9.4; SAS Institute Inc., Cary, NC, USA), turkeys were grouped into three groups based on their non-aggressive peck ratios, including a high pecking group, a moderate pecking group and a low pecking group. This was done for every room on the 14th week of age prior to any treatment. Further statistical analyses were conducted to verify that pecking activity differed significantly among clusters. Fresh meat quality attributes were then compared among clusters to assess possible relationships between meat quality attributes and a turkey’s peck ratio.

### 2.5. Harvesting

To determine the processing order for the birds in each room, each bird was marked with a color (red, orange, yellow, green, blue, purple or pink) using a non-toxic livestock marker (Prima Tech Marking Stick, Neogen Corp., Lansing, MI, USA) the day before harvesting. Red or orange colors were given to the two birds that wore accelerometers in each room and the rest of the birds in each room were randomly assigned a color. At 12 h prior to harvesting, feed was taken from each room to ensure that the birds were fasted before evisceration. Drinking water was left in the rooms and was provided up until harvest.

At 15 week of age, turkeys were transported to the Purdue Boilermaker Butcher Block for harvesting and sample processing. Birds were slaughtered under standard conditions of electrical stunning, bleeding for 120 s, scalding at 60 °C (3–4 min) and feather removal in a rotary drum plucker. Eviscerated carcasses were weighed (Hot Carcass Weight), and initial pH measurements were taken before carcasses were air-chilled in a 2 °C carcass cooler breast-side up for 24 h After carcasses were chilled, ultimate pH measurements were taken (specific procedures described below). Breast meat (*M. Pectoralis major*) was cut from the right side of each carcass and weighed. The right breast was cut into 2.54 cm slices to determine water-holding capacity, cook loss, Warner-Bratzler shear force, instrumental color, proximate analysis and fatty acid composition. Following sample collection, slices were individually vacuum packaged and then frozen at −40 °C until further use.

### 2.6. Feather Retention Force

Prior to feather removal, the feather release force was taken for each bird after scalding. Three mature feathers from the tail were pulled out individually with a hemostat attached to an FG-3008 digital force gauge (Nidec-Shimpo Corporation, Glandale Heights, IL, USA) to measure the force required to pull out each feather (adapted from Pool [30]). The three recorded measures were then averaged for each bird.

### 2.7. pH Measurements

The pH values were measured in duplicate from two randomly selected locations on the ventral side of the right breast of each carcass approximately 20 min post-mortem. These same locations were used for the ultimate pH measurement taken after 24 h of air chilling at 2 °C. Measurements were taken using a calibrated meat pH probe that was directly inserted approximately 1 cm into the muscle tissue (HANNA HI 99163, Hanna Instrument, Inc., Warner, NH, USA).

### 2.8. Instrumental Color Measurements

After the right breast muscle had been sliced into 2.54 cm sections, the slices designated for the Warner-Bratzler shear force test were measured in three randomly selected locations using a Hunter MiniScan EZ colorimeter (Hunter, Reston, VA, USA). The setting for the illuminant was D_65_ source, and the observer was at the standard 10°. The CIE lightness (L*), redness (a*) and yellowness (b*) values were recorded. Following color measurements, samples were frozen at −40 °C until further use.

### 2.9. Drip Loss

Drip loss was measured according to the Honikel drip loss protocol [31] as described by Kim et al. [32] and expressed as percent difference between the initial sample weight and 24 h sample weight, and initial sample weight and 48 h sample weight after hanging in a plastic storage container at 4 °C.

### 2.10. Cook Loss and Warner-Bratzler Shear Force

To determine cook loss, raw samples were individually weighed to measure initial weight. Samples were then cooked in an 80 °C water bath until internal temperatures reached 71 °C, which was monitored using a T-type thermocouple (Omega Engineering, Stamford, CT, USA) connected to an OctTemp 2000 data logger (Madge Tech, Inc., Warner, NH, USA). Once samples reached 71 °C, the samples were immediately removed from the water bath, cooled under refrigeration, and then weighed. Cook loss was expressed as the percent change between the initial and final weight of the samples.

Once completely cooled, six 1 cm × 1 cm slices were cut from each sample parallel to fiber direction to be used for the Warner-Bratzler shear force test. Slices were then sheared perpendicularly to fiber direction using a TA-XT Plus Texture Analyzer (Stable Micro System Ltd., Surrey, UK) with the Warner-Bratzler shear attachment. Test speeds were set at 2 mm/s and peak shear force in Newtons per slice was then averaged to calculate an average shear force value for each sample.

### 2.11. Proximate Analysis

Proximate analysis of breast meat samples was conducted using the AOAC guidelines [33]. Two birds from each pen that were closest to the pen average live body weight were chosen for sampling. Moisture was determined in triplicate measurements using the oven air-drying method at 105 °C and weighing the samples before and after drying. Ash was measured in triplicate by combusting dried samples in a 580 °C muffle furnace and weighing the samples before and after ashing. Nitrogen was measured in duplicate using the Dumas combustion method and then multiplied by 6.25 to determine crude protein concentration (Leco, St. Joseph, MI, USA). Lipid content was determined using the methods by Folch et al. [34].

### 2.12. Fatty Acid Composition

Intramuscular lipids were extracted in duplicate from powdered samples using the method described by Folch et al. [34] with modifications described by Shin and Ajuwon [35]. Only two birds from each pen that were closest to pen average live body weight were chosen to sample for this portion. Fatty acid methyl esters (FAME) were prepared from the extracted lipids by adding sodium methoxide to methanol. The FAME were analyzed using a gas chromatograph (Varian CP 3900) equipped with a 105 m Rtx-2330 (Restek) fused silica capillary CG column (0.22 mm ID and 0.20 µm df). Helium was used as the carrier gas with a flow rate of 40 mL/min. Injector and detector temperatures were at 260 °C. Injection volume was set at 1 µL with a 50:1 split injection. The column oven temperatures were increased from 140 °C to 180 °C at a rate of 8 °C/min, from 180 °C to 260 °C at a rate of 5 °C/min, and then held at 260 °C for 15 min. The fatty acids were identified by comparing them to a retention time of a known standard (Supelco 37 components FAME Mix, Sigma-Aldrich, St. Louis, MO, USA) and the peak area of the fatty acid detected was expressed as a percent of the total peak area.

### 2.13. Statistical Analysis

Treatment (HS, CON, IC; 1 week prior to harvest) effects on meat quality were analyzed using PROC MIXED (SAS version 9.4; SAS Institute Inc., Cary, NC, USA) for all data except shear force. Body weight was included as a covariate; room nested within trial was included as a random effect. Tukey’s test for multiple comparisons was used to determine post hoc differences among treatment groups. Normality of data was verified by examining qq plots and plots of studentized residuals. Shear force values were log-transformed to meet normality assumptions and were analyzed using PROC GLIMMIX, using the link=log function and ilink option. To examine the relationship between social tension based on non-aggressive pecking behavior and meat quality, the FASTCLUS procedure of SAS version 9.4 (SAS Institute Inc., Cary, NC, USA) was used to separate the birds into three groups based on their non-aggressive pecking frequency value. The GLM procedure was then used to verify that the pecking values were different among clusters. All analyses used post hoc Tukey tests for multiple comparisons, and body weight was included as a covariate. The MIXED procedure was used to compare the meat quality parameters among the resulting clusters, and room and trial were included as random effects. Significance was set at *p* < 0.05 for all analyses.

## 3. Results

Treatment had a significant effect on percent moisture per sample (*p* = 0.04), as birds subjected to HS had the greatest moisture, and no differences were found between CON and IC (Table 7). Treatment also had a significant effect on percent protein (*p* = 0.03); birds subjected to HS had lower percent protein compared to CON (*p* = 0.04), and IC birds were intermediate. Percent fat and ash were not different among treatment groups (*p* = 0.82 and *p* = 0.10 respectively; Table 7).

Fatty acid composition (g/100 g) of intramuscular lipids in the turkey meat did not differ among treatment groups for all 38 fatty acids tested; nor for the percentage of total saturated, monounsaturated, and polyunsaturated fatty acids; and nor did the ratio of saturated to unsaturated fatty acids (*p* > 0.05; Table 8). However, Caproic (C6:0) and Heptadecanoic (C17:0) tended to differ (*p* = 0.10 and *p* = 0.07, respectfully) among treatment groups.

Treatment had a significant effect on initial pH_20min_ values (*p* = 0.01; Table 9) as birds subjected to IC had higher pH values compared to CON, and HS birds were intermediate (Table 9). Treatment also had a significant effect on CIE L*, or lightness (*p* = 0.04); IC birds had lower CIE L* compared to HS and CON, which did not differ from each other. Lastly, treatment had a significant effect on shear force (*p* = 0.03) as birds subjected to IC before slaughter had higher shear force values compared to CON (*p* = 0.03; Table 9), and HS birds were intermediate.

Nonaggressive pecking frequencies were categorized into three clusters by non-aggressive interaction scores (±SD) of: high (cluster 3; 0.92 ± 0.09), medium (cluster 1; 0.56 ± 0.11), or low (cluster 2; 0.23 ± 0.08). The higher the value, the more pecking was given to others rather than received. Instances of aggressive pecking behaviors were so infrequent (<1 interaction per bird per hour) that there was not enough data to use it as a means for comparing bird behavior to meat quality measurements, thus only non-aggressive pecking behaviors were evaluated. None of the meat quality parameters measured, including feather retention force, differed among clusters (*p* > 0.05; Table 10.).

## 4. Discussion

Some aspects of turkey meat quality were affected by a mild heat stress and vaccination before slaughter. Specifically, turkeys in the HS group had lower percentages of protein, higher pH and greater shear force values compared to CON, and turkeys in the IC group had higher pH and higher shear force values compared to the CON and HS groups. A limitation of this study was that the birds were subjected to all three treatment conditions which may have led to possible carryover effects; however, there is limited research available on the influence of an immune challenge on turkey meat quality, and on the possible carryover effects of heat stress and an immune challenge. To reduce possible carryover effects on final meat quality, we waited two weeks between treatments. Further research is needed to examine interactions among heat stress and an immune challenge on turkey meat quality. Furthermore, additional analysis of more meat quality traits, such as lipid oxidation and myoglobin redox instability, may provide additional insights.

When comparing turkey meat quality to that reported in other studies, the main fatty acids found in the breast muscles were Palmitic (C16:0), Stearic (C18:0), Oleic (C18:1n9c), Linoleic (C18:2n6c), and Arachidonic (C20:4n6) acid, which is consistent with the findings of Baggio et al. [36] of the most abundant fatty acids being C18:2n6, C18:1n9, C16:0, C18:0, and C20:4n6. We were unable to find any similar studies of fatty acid composition of heat-stressed or immune-challenged turkeys to compare to. Wong et al. [37] reported higher amounts of Palmitic acid (C16:0) and total percent of polyunsaturated fatty acids, but very similar amounts of Linoleic acid (C18:2n6c) in their raw turkey composite for light meat compared to the results in the present study. Polyunsaturated fatty acids for all three treatment groups present in this study were similar to those found in a study by Wong et al. [37].

The IC and CON birds were very similar in proximate composition, with only the HS group being significantly different for protein and moisture content. Protein content was significantly lower for the HS group which correlated well with the higher moisture content for that group of birds. Very similar amounts of percent moisture were found in other studies of turkey meat by Wong et al. [38] and Paleari et al. [39] with 74.4% and 74.8%, respectively, which are most similar to the HS group of birds, and much higher than the percent moisture found in the CON and IC birds. A study on broiler meat by Zhang et al. [39] also reported higher moisture content and lower protein content in the breast muscle, similarly to what we report here. Higher ambient temperatures significantly decrease body protein content by changing protein metabolism, decreasing protein synthesis, and increasing protein breakdown [39,40,41]. It has also been demonstrated that heat stress lowers protein synthesis by changing ribosomal gene transcription [42,43]. Therefore, the decreased protein content in the HS birds could be attributed to lower ribosomal capacity and a decreased rate of protein synthesis resulting in the reduction of protein deposition; however, further analysis would need to be completed to determine this conclusively.

The amount of water and how well it is distributed within muscles may affect the visual appearance of meat, but also affects the tenderness and juiciness. A similar study with heat-stressed and control groups of turkeys by McKee and Sams [44] showed CIE L* values at 53, which is much higher than any of the mean CIE L* values for all three treatments in the present study. However, McKee and Sams [44] reported lower CIE L* values for their control group compared to our CON and HS groups, but very similar ones to our IC group. The study by McKee and Sams [44] presented their birds with a much longer heat stress period, and their turkeys were 17 weeks of age compared to ours at 15 weeks of age. McKee and Sams [44] also reported that acute and chronic heat stress can cause poor water-holding properties in meat. Stress conditions cause a high metabolic rate during rigor mortis that causes more protein denaturation that prevents the protein’s ability to bind water as effectively [45]. The turkey breasts sampled did not exhibit any typical pale, soft, and exudative (PSE) conditions for any of the treatments here that are usually found with porcine muscle under similar conditions. Findings by Barbut [46] and Owens et al. [47] suggested that higher CIE L* values (>51–53) were associated with PSE meats and paler coloring, a changed texture, and poor water holding capacity. However, in the present study, the HS and IC groups with the higher pH values and lower CIE L* values showed poorer water holding capacity in drip loss, and tougher texture measured by shear force. Cook loss values did show a trend of poor water holding capacity with paler colored meat. The CON and HS groups had higher CIE L* values of 51.5 and lower pH values indicating the start of PSE-like conditions, but not enough to cause a significant issue with the final product quality. It is interesting to note that the CON birds had lower initial pH_20min_ values compared to the two stress treatments since lower pH values are usually associated with PSE-like conditions in swine and poultry [48]. A study by Çelen et al. [49] showed that the initial pH of normal turkey breast muscle is about 6.20, whereas the pH_20min_ in our CON group was as low as 6.1 and our IC group as high as 6.25. Shear force values, or tenderness, were observed to be significantly higher for the HS and IC groups. The increased shear force values under the stress conditions were closely related to results reported in several studies with other food animal species.

The higher pH_20min_ values and shear force values for the IC birds correspond with results of a study on broiler meat by Mellor et al. [50] where they found birds with a higher muscle glycogen content at slaughter to have lower final muscle pH and lower shear force values than the other birds. In the present study, our IC birds probably had less glycogen content at slaughter and therefore had a final muscle pH_24h_ that was approaching significance and significantly higher shear force values. Weary et al. [51] concluded that animals who consume less feed while displaying sickness behaviors will use up stored glycogen for energy when stressed, leaving less glycogen to be converted into lactic acid during the muscle to meat conversion and thus higher pH and shear force values. Another explanation for the increased shear force values in stressed turkeys could be due to the excessive amounts of reactive oxygen species (ROS) that the body can generate that will lead to oxidation of the sarcoplasmic and myofibrillar proteins, and ultimately reduce the proteins’ solubility and ability to bind water [52]. Heat stress has been suggested to be an environmental factor that increases the production of ROS [53]. This inability to bind water leads to increases in drip loss and cook loss, and reduces the water-holding capacity, juiciness and tenderness of meat [52].

Feather retention force was variable but did not differ among treatment groups. The force required to pull out the feathers is also highly dependent on scald tank temperature and time in the tank, with scald temperature being the most critical component [30]. Pool et al. [30] also noted that higher temperatures make the feathers easier to pull out, but also risk the possibility of prematurely cooking the meat or dehydrating it. The scald tank temperature in this study was kept at 60 °C, which is consistent with the standard methods of poultry slaughter. To the best of our knowledge, there are no published studies examining feather retention force of turkeys.

Our results did not support our hypothesis that individual pecking behavior influenced turkey meat quality. Due to low frequency of aggressive interactions, we were unable to examine the relationship between aggression and meat quality, and used the frequency of non-aggressive pecking instead. We did not examine whether pecking behavior was consistent among turkeys, and further research is needed to fully examine the characteristics of non-aggressive pecking of turkeys. However, research with laying hens demonstrated that some birds remained consistent in feather pecking behavior [54] and that consistent behavioral patterns can be related to certain aspects of meat quality (e.g., bulls) [55,56]. Considering none of the traits or meat quality parameters measured were statistically significant between cluster groups and the fact that there is very little information comparing poultry pecking behaviors and meat quality, there is limited discussion possible on this subject. However, there is enough information on this topic in cattle and some poultry species that we can still compare results. A similar study analyzing the approach-avoidance behaviors in humans also used a cluster analysis to categorize people into three qualitatively different groups based on their behaviors [57]. This method of cluster analysis can reveal statistically reliable and distinct groups; this is why we decided that using a cluster analysis on this present study was the best way to categorize the birds into three distinct groups based on pecking frequency. Living animals form dominance hierarchies as a result of a number of dyadic dominance relationships and social interactions [58]. There are several hypotheses as to how these relationships are developed; however, pecking behaviors (aggressive or not) in poultry species are a good indicator of the possible social hierarchy at large [28].

There were no significant differences among clusters for carcass traits; however, there was a noticeable value differences between cluster 1 and the other two for feather retention force. Birds in the middle of the social index had higher feather retention scores indicating some possible signs of stress with tightening of the smooth muscle that surrounds the hair follicle [24]. No significant differences in hot carcass weight indicated that there was no noticeable difference in size between the clustered groups. This is a dissimilar result to that of studies with cattle where the lightest and heaviest in the pens experienced the most social stress [59]. The difference with these birds could be due to their young age and ample amount of room in the pen to feed without having to compete for resources.

A previous study by Andrighetto et al. [60] revealed that social stress not only affects muscle color, firmness, and water holding capacity of the meat, but also tenderness. Considering none of our meat quality parameters, including pH, color, water holding capacity, and tenderness, were significantly different among clusters, we can say that the social stress related to non-aggressive pecking was not significant enough to influence meat quality. However, prior to post hoc analysis, there was a trend for significant differences (*p*-values < 0.10) for L* (lightness) and drip loss which would be similar to the results of Andrighetto’s study.

It appears that stress conditions prior to slaughter have an important role in subsequent turkey meat quality characteristics. Better control of the environment of the birds prior to slaughter and better ways of dealing with behavioral changes and individual responses after vaccines should be considered, including sickness behaviors, in order to minimize color and toughness problems. Turkeys in this study were harvested one week after the vaccine was administered, and a longer period between vaccine administration and harvesting would help clarify some of the observed results. Considering there are no previous studies on the impacts of the hemorrhagic enteritis vaccine on meat quality, it is difficult to compare our results to others. However, immune stressing birds prior to slaughter may possibly cause them to exhibit sickness behaviors and to use up their glycogen energy stores, which can have detrimental effects on meat quality [51].

Poultry scientists are interested in knowing how dominant and social interactions are associated with well-being and production characteristics such as meat quality [11]. Thus, detailing these relationships and measuring the outcomes are needed to understand the associations between these characteristics. In addition, multiple factors such as genetics, management systems, and physical and social environments are open to scrutiny to determine whether it is behavioral or other issues that cause the differences in production outcomes. Further research examining turkey behavioral traits and welfare will be valuable in identifying factors that influence turkey meat quality. Moreover, detailing the behavioral traits of turkeys at an older age and in a denser population may be more applicable to large scale production, and offer a better insight into the relationship between individual behavioral characteristics and meat quality.

## 5. Conclusions

The heat stress treatment did not appear to have as detrimental an effect as expected on meat quality, while the immune-challenged birds did show more of a negative effect. However, differences between treatments and controls were varied amongst measurements. Our data support the use of an appropriate recovery time after a vaccine is given before the birds are slaughtered, and that withdrawal periods are important not just for the safety of humans, but also for better quality meat products. Furthermore, studies with a more extreme heat stress may cause more of an effect on meat quality and other carcass traits such as feather retention force. Finally, the turkeys’ tendency to perform and receive non-aggressive pecks did not seem to influence the carcass traits or meat quality attributes tested in this study.

## Figures and Tables

**Table 1 foods-11-02203-t001:** Schedule and order of treatments imposed on turkeys in each room: Heat Stress (HS), Immune Challenge (IC), or Control (neither HS or IC; CON). The experiment was run in two trials, with two to three replicate rooms per treatment in each trial (5 replicates/treatment total). Treatments were applied 2 weeks apart, when turkeys were 10, 12 and 14 weeks of age.

Bird Age (Weeks)	Trial 1 (3 Rooms) Trial 2 (2 Rooms)	Trial 1 (3 Rooms) Trial 2 (2 Rooms)	Trial 1 (2 Rooms) Trial 2 (3 Rooms)
10	IC	CON	HS
12	CON	HS	IC
14	HS	IC	CON

**Table 2 foods-11-02203-t002:** Average (±SD) humidity and temperature ranges during Trial 1 and Trial 2 at each age, recorded 10 min prior to heating the rooms. Target temperature ranges were derived from Xin and Harmon [26].

	Humidity (%) Pre Heat Stress Treatment	Temperature (°C) Pre-Heat Stress Treatment	Target Temperature (°C) Range for Heat Stress	Bird Age (Weeks)
Trial 1	55.0 ± 1.0	17.4 ± 0.1	27.8–29.4	10
	16.5 ± 1.5	17.9 ± 0.2	29.4–30.6	12
	24.5 ± 1.5	17.7 ± 0.1	29.4–30.6	14
Trial 2	15.9 ± 0.9	18.0 ± 0.3	29.4–30.6	10
	51.5 ± 0.4	17.3 ± 0.1	27.8–29.4	12
	20.0 ± 1.3	18.3 ± 0.1	29.4–30.6	14

**Table 3 foods-11-02203-t003:** Cloacal temperatures (average ± SD) of turkeys at each age period (10 week, 12 week and 14 week) 10 min before (pre heat stress treatment), after birds had experienced peak temperatures for 2 h (during peak heating), and 2 h after the imposed heat stress (post heat stress treatment).

Cloacal Temperature (°C) Pre-Heat Stress Treatment	Cloacal Temperature (°C) during Peak Heat Stress Treatment	Post Heat Stress Treatment (°C)	Bird Age (Weeks)
40.6 ± 0.3	41.1 ± 0.3	40.6 ± 0.4	10
40.6 ± 0.1	41.4 ± 0.3	40.6 ± 0.3	12
40.0 ± 0.4	41.4 ± 0.4	40.0 ± 0.4	14

**Table 4 foods-11-02203-t004:** Timeline of imposed heat stress and immune challenge treatments on the treatment day.

	Time of Day				
Stress Treatment	0700	0900	1100	1300	1500
Heat Stress Room	Nothing	Start heating to peak temperature	Peak temperature achieved	Temperature declining to normal	Pre-heat stress temperature achieved
Immune Challenge Room	Water deprivation begins	Vaccine added to drinking water	Vaccine water removed and non-vaccine water restored		

**Table 5 foods-11-02203-t005:** Bird role in behavior observed continuously for 15 min at 1300 and 1600 h two days before the treatments were imposed.

Bird Role	Description
Instigator	The bird who is responsible for starting the interaction: the first bird to peck, chase, threat, etc. another bird
Recipient	The bird who is receiving the action that the instigator bird started

**Table 6 foods-11-02203-t006:** Ethogram of turkey behavior observed continuously for 15 min at 1300 and 1600 h two days before the treatments were imposed.

Behavior	Description
Aggressive Peck	Pecking at another bird on the face, head or neck repeatedly: usually, a forceful downward peck that results in the recipient moving away. Pecking or grabbing on to the neck or snood of another bird may be exhibited
Feather Pecking	Using beak and extending neck to peck at the feathers of another bird: feathers are sometimes pulled, but usually not pulled out
Beak Pecking	Using beak to gently peck at another bird’s beak, neck, or face
Chase	Running (or chasing) towards another bird in an aggressive manner (neck and head are stretched out in a threatening posture and feathers are erect)
Threat	Head is raised in front of another bird and this is sometimes accompanied with raising of the feathers of the neck (neck and head are stretched out in a threatening posture). The bird that is doing the threatening action is considered the instigator
Avoidance	Walking or running away from another bird (also includes moving out of the way of another bird), usually accompanied with lowering of the head. The bird that is doing the avoiding is considered the recipient

**Table 7 foods-11-02203-t007:** Proximate composition of turkey meat subjected to environmental stress 1 week prior to harvest.

	Stress Treatment ^1^				
Proximate Composition ^2^	CON	HS	IC	SEM	Significance of *p*-Value
Moisture (%)	72.1 ^b^	75.2 ^a^	71.9 ^b^	0.94	0.04
Protein (%)	22.0 ^a^	18.9 ^b^	21.6 ^ab^	0.16	0.03
Fat (%)	1.72	1.89	2.00	0.31	0.82
Ash (%)	4.18	4.00	4.55	0.93	0.10

^1^ Environmental treatment 1 week prior to harvest: Control (CON) = no treatment, Heat Stress (HS) = target room temperature of 29.4–30.6 °C for 120 min each day, Immune Challenge (IC) = live-virus hemorrhagic enteritis vaccine given via drinking water for 120 min each day, SEM = standard error of the mean. ^2^ Mean and standard error of triplicate measurements of moisture and ash, and duplicate measurements of protein; fat was calculated as the remaining value to reach 100%. ^ab^ Means lacking a common superscript differ due to stress treatment (*p* < 0.05).

**Table 8 foods-11-02203-t008:** Fatty acid composition (g/100 g) of intramuscular lipids in turkey meat subjected to environmental stress 1 week prior to harvest.

		Stress Treatment ^1^				
Fatty Acid		CON	HS	IC	SEM	Significance of *p*-Value
C4:0	Butyric	1.91	1.67	1.67	0.29	0.80
C6:0	Caproic	0.54	0.89	0.74	0.11	0.10
C8:0	Caprylic	0.30	0.27	0.22	0.03	0.27
C10:0	Capric	0.12	0.11	0.10	0.01	0.47
C11:0	Undecanoic	0.04	0.04	0.05	0.01	0.45
C12:0	Lauric	2.07	1.51	1.34	0.42	0.46
C13:0	Tridecanoic	0.05	0.06	0.04	0.01	0.72
C14:0	Myristic	0.41	0.46	0.45	0.03	0.45
C14:1	Myristoleic	0.08	0.10	0.17	0.03	0.17
C15:0	Pentadecanoic	0.18	0.18	0.20	0.03	0.95
C15:1	cis-10-Pentadecanoic	0.07	0.09	0.05	0.01	0.23
C16:0	Palmitic	16.6	16.9	17.3	0.31	0.30
C16:1	Palmitoleic	0.88	1.17	1.10	0.12	0.21
C17:0	Heptadecanoic	0.20	0.22	0.18	0.01	0.07
C17:1	cis-10-Heptadecanoic	0.07	0.17	0.14	0.04	0.28
C18:0	Stearic	10.1	9.57	10.02	0.40	0.50
C18:1n9t	Elaidic	0.14	0.14	0.15	0.04	0.96
C18:1n9c	Oleic	13.3	14.7	14.8	0.78	0.33
C18:2n6t	Linolelaidic	0.04	0.07	0.09	0.03	0.34
C18:2n6c	Linoleic	23.3	23.4	23.2	1.12	0.99
C20:0	Arachidic	0.07	0.07	0.06	0.01	0.63
C18:3n6	γ-Linoleic	0.24	0.27	0.25	0.02	0.57
C20:1n9	cis-11-Eicosenoic	0.12	0.14	0.15	0.03	0.70
C18:3n3	α-Linoleic	1.32	1.41	1.31	0.12	0.82
C21:0	Heneicosanoic	0.17	0.14	0.08	0.05	0.45
C20:2	cis-11,14-Eicosadienoic	0.53	0.50	0.49	0.03	0.50
C22:0	Behenic	0.11	0.08	0.07	0.02	0.21
C20:3n6	cis-8,11,14-Eicosatrienoic	0.71	0.51	1.64	0.47	0.23
C22:1n9	Erucic	1.31	0.88	0.37	0.42	0.33
C20:3n3	cis-11,14,17-Eicosatrienoic	1.19	1.85	1.12	0.51	0.23
C23:0	Tricosanoic	0.09	0.32	0.02	0.19	0.29
C20:4n6	Arachidonic	7.76	6.52	7.47	0.55	0.28
C22:2	cis-13,16-Docosadienoic	0.05	0.06	0.03	0.01	0.33
C24:0	Lignoceric	0.04	0.03	0.02	0.01	0.14
C20:5n3	cis-5,8,11,14,17-Eicosapentaenoic	0.24	0.19	0.24	0.02	0.10
C24:1n9	Nervonic	0.03	0.03	0.03	0.01	0.74
C22:6n3	cis-4,7,10,13,16,19-Docosahexaenoic	0.90	0.68	0.85	0.09	0.21
Saturated (%)		33.8	32.8	32.7	0.86	0.62
Monounsaturated (%)		15.9	17.4	17.1	0.84	0.45
Polyunsaturated (%)		35.6	35.2	36.8	1.18	0.64
Saturated:Unsaturated		0.66	0.63	0.61	0.03	0.59

^1^ Environmental treatment 1 week prior to harvest: Control (CON) = no treatment, Heat Stress (HS) = target room temperature of 29.4–30.6 °C for 120 min each day, Immune Challenge (IC) = live-virus hemorrhagic enteritis vaccine given via drinking water for 120 min each day, SEM = standard error of the mean.

**Table 9 foods-11-02203-t009:** Harvesting and meat quality attributes of turkey meat subjected to environmental stress 1 week prior to harvest.

	Stress Treatment ^1^				
Trait	CON	HS	IC	SEM	Significance of *p*-Value
Feather Retention Force (N)	21.9	25.9	21.4	3.26	0.57
Hot Carcass Weight (kg)	8.57	8.57	8.53	0.05	0.73
pH_20min_	6.10 ^a^	6.17 ^ab^	6.25 ^b^	0.03	0.01
pH_24h_	5.65	5.61	5.81	0.08	0.16
CIE L* (lightness)	51.5 ^a^	51.5 ^a^	50.2 ^b^	0.41	0.04
CIE a* (redness)	12.1	12.2	12.8	0.35	0.34
CIE b* (yellowness)	11.7	12.0	12.4	0.69	0.75
Drip Loss_24h_ (%)	0.52	0.57	0.59	0.12	0.92
Drip Loss_48h_ (%)	0.92	0.94	0.98	0.16	0.97
Cook Loss (%)	11.8	12.5	11.5	0.32	0.07
Shear Force (N)	17.7 ^a^	18.3 ^ab^	21.6 ^b^	1.12	0.03

^1^ Environmental treatment 1 week prior to harvest: Control (CON) = no treatment, Heat Stress (HS) = target room temperature of 29.4–30.6 °C for 120 min each day, Immune Challenge (IC) = live-virus hemorrhagic enteritis vaccine given via drinking water for 120 min each day, SEM = standard error of the mean. ^ab^ Means lacking a common superscript differ due to stress treatment (*p* < 0.05).

**Table 10 foods-11-02203-t010:** Meat quality measurements of turkeys divided into clusters based on non-aggressive pecking frequency.

	Cluster of Non-Aggressive Pecking ^1^				
Trait	Low	Medium	High	SEM	Significance of *p*-Value
Feather Retention Force (N)	24.6	21.6	21.6	2.26	0.17
Hot Carcass Weight (kg)	18.5	19.0	19.0	0.22	0.13
pH_20min_	6.19	6.17	6.18	0.02	0.61
pH_24h_	5.72	5.70	5.72	0.04	0.63
CIE L* (lightness)	50.9	50.9	51.0	0.31	0.93
CIE a* (redness)	12.3	12.5	12.3	0.28	0.62
CIE b* (yellowness)	11.8	12.0	12.1	0.45	0.42
Shear Force (N)	18.9	20.9	18.3	1.17	0.31

^1^ Non-aggressive pecking frequencies: (All non-aggressive pecks given + 1)/(All non-aggressive pecks received + 1). Low cluster = 0.23, Medium cluster = 0.56, and High cluster = 0.92, SEM = standard error of the mean.

## Data Availability

The data are available from the corresponding author.

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
