# Peer review of "Heat Stress and an Immune Challenge Influence Turkey Meat Quality, but Conspecific-Directed Pecking Behavior Does Not"

_foods, 2022, doi:10.3390/foods11152203_

Round 1
Reviewer 1 Report
It is an interesting study. The results could be useful for the Turkey Industry. I have made some suggestions:
Title: Is "social behavior" the best terminology in this case? Aggressive behavior, perhaps?
L20-21 - It is hard to understand this sentence. Little difference between HS and IC groups ? Or between HC/IC and the control group? What were these differences? A better quality or a worse quality?
The introduction is appropriate.
L117 - Why was a crossover design chosen? I think some carryover effects may affect your data. Please provide a rationale for this.
I had trouble understanding Table 1; I really tried, but I do not understand it. There is something missing, for example, a left column. If I am correct, perhaps you should add time as a column and the different trials as rows... Please add a left column to lead the table.
L331 - "were"
Table 7 - Please define SEM - I think it is a standard error, but the table needs to stand for itself.
Why do not you give SE or SD for each value?
Table 10 - Since you did not find a correlation between behavior and meat quality, should you keep this in the title? The title is quite concise. When I read it, I was pretty sure a correlation was found.
Another point: you did not really test for associations/correlations, but for differences. Perhaps the verb in the title could be changed for clarity.
Author Response
Please see the attachment and revised manuscript that has addressed these comments.

Author Response
Please see the attachment and the updated manuscript

Reviewer 3 Report
1. Line 20; Minor differences in carcass and meat quality…please state what quality parameters?
2. Line 234. How long for scalding?
3. Section 3.9. Drip loss or WHC?
4. Line 270-271. until internal temperatures 270 reached 71° C… how long did it hold at this temperature?
5. Line 289-291. Carbohydrate composition was assumed to be approximately 0%; thus, lipid concentration was determined as 100% - (% moisture + % protein + % ash). Why? Total lipid content can be obtained from Folch method (Line 294).
6. Lipid oxidation and myoglobin redox instability among the treatments should be analyzed?
Author Response
Please see the attachment and the updated manuscript for corrections

Reviewer 4 Report
- Abstract lacks the results of the study, it must stands by it self, therefore, the authors must give the reader a clear idea about the study in the abstract (i.e., objectives, materials and methods, results and conclusion), please add some results
- At the end of the abstract, please add a conclusion statement
- Although it is a challenge study, the period of one week is not enough to draw a clear idea about the results, please clarify
- In the introduction, authors focused on giving too much information on the social behavior stress with a little information regarding heat stress, please balance between the two factors
- In the materials and methods, please add some information about the diet used in the study
- Since there were significant differences were detected in the fatty acid composition (Table 8), please omit this table and add it as a supplement
- In the discussion section, too much literature is presented, please try to cut it and focus on discussing the obtained results
- Rewrite the conclusion to reflect the results of the study
Author Response

(The authors gave the same response as above.)

Round 2
Reviewer 3 Report
All points raised by reviewers were carefully addressed and answered point-by-point. So, it can be accepted.
Reviewer 4 Report
The authors have addressed all comments showed in the first version of this manuscript. Thanks